# Spousal Violence and Contraceptive Use among Married Afghan Women in a Nationally Representative Sample

**DOI:** 10.3390/ijerph19169783

**Published:** 2022-08-09

**Authors:** Sahra Ibrahimi, Julia R. Steinberg

**Affiliations:** Department of Family Science, School of Public Health, University of Maryland, 2242 Valley Drive, College Park, MD 20742, USA

**Keywords:** spousal violence and types, contraceptive use and types, demographic health survey, married Afghan women, the Taliban administration

## Abstract

Objective: Afghanistan is one of the countries with the highest prevalence of spousal violence (56%) and a low prevalence of contraceptive use (23%), yet there is no study assessing how spousal violence is related to contraceptive use, and what methods are most used by women. Therefore, this study examined the association between the number of types of spousal violence and contraceptive use. Method: Using data from 18,985 Afghan married women, aged 15 to 49, who responded to the 2015 Afghanistan Demographic and Health Survey, the current contraceptive method was grouped into five categories: male-involved methods, pills, injectables, long-acting reversible contraception, female sterilization, and Lactation Amenorrhea Method. The number of types of spousal violence in the past 12 months was categorized as none, one type, or two or more types, based on women’s experiences with verbal, physical, and sexual violence. For analysis, binary and multinomial logistic regression were used. Results: After adjusting for the covariates, the experience of any spousal violence was associated with contraception use (adjusted odds ratio (aOR) = 1.93, 95% CI: 1.64–2.27, *p* = 0.0001). Among those using contraception, experiencing two or three types of spousal violence was associated with using pills (adjusted risk ratio (aRRR) = 2.12, 95% CI: 1.63–2.77, *p* = 0.0001), injections (aRRR = 1.75, 95% CI: 1.26–2.41, *p* = 0.001), and LAM (aRRR = 3.27, 95% CI: 2.05–5.20, *p* = 0.0001), compared to male-involved methods. Conclusions: The findings of this study may inform policymakers and program implementers in designing interventions to address the pervasive problem of violence against women, and make pills and injectables more accessible to Afghan women, since these methods are under women’s control and more often used in Afghanistan.

## 1. Introduction

Since 1973, the people of Afghanistan have been enduring a continuous war [1]. The war and conflict have created structural violence that can shape people’s self-conceptions, behaviors, and practices [2]. War and the experience of structural violence are associated with the perpetration of intimate partner violence (IPV), and this association is mediated by psychopathology, including posttraumatic stress disorder (PTSD) and depression [3,4]. War is linked to PTSD diagnosis, and a trait of PTSD is the loss of self-regulatory and impulse control, which can induce and exacerbate IPV [3]. War also robs people of opportunities for education and employment [5]. In a patriarchal society such as Afghanistan, men’s diminished opportunities for employment and providing for their families may challenge the norm of hegemonic masculinity, which refers to a set of beliefs and dominating practices that define an “ideal” manhood [6]. Jewkes and Morrell (2010), in their study of South Africa, assert that male toughness, the possession of multiple sexual partners, non-condom use, and the perpetration of violence stem from hegemonic masculinity [7].

The hegemonic masculinity, patriarchal culture, and lack of law to protect women’s rights in Afghanistan have resulted in a society where women have less rights than men [5]. After the civil war in 1996, when the Taliban took over Afghanistan, the Taliban ruled by their unique model of Islam, which was considered the darkest era for Afghan women in the country [8]. Girls and women were not allowed to get an education [9]. Women could also not step outside their home without being escorted by a male relative and covered with the head-to-toe burqa [10]. After the terrorist attacks on September 11, 2001 in the United States (U.S.), the U.S. defeated the Taliban and helped Afghanistan establish a government [8]. Although the Taliban were no longer in power, they continued to orchestrate attacks on schools and threaten women’s rights [10]. 

In August 2009, the United Nations Assistance Mission in Afghanistan (UNAMA) helped the Afghanistan government ratify the Elimination of Violence Against Women Law (EVAWL) in order to establish women’s equal rights and protection against violence [11]. UNAMA conducted research from March 2010 to September 2011 to assess the impact of EVAWL. UNAMA found that many cases of domestic violence and serious crimes were prosecuted using the Penal Code or the Sharia law instead of the EVAW law [11]. As a result, sometimes, perpetrators were exonerated or convicted with lighter sentences, and women victims themselves were accused of “moral crimes” [11].

Despite the international efforts to improve women’s condition in Afghanistan, domestic violence remains a major public health concern. According to Afghanistan’s 2015 Demographic Health Survey, a large nationally representative sample, more than 56% of ever-married Afghan women between ages 15 to 49 years reported spousal violence in 2015 [12]. The ubiquitous violence against women in Afghanistan has been acknowledged by many national and international agencies. A report by the World Health Organization (WHO) in 2011 showed that 15% of girls were married before the age of 15, and 46% before the age of 18 [13]. About 92% of women believed that a husband is justified in beating his wife [13]. The majority (88%) of Afghan women have no jobs and are financially dependent on men [12]. These inequalities position women to be more likely to experience violence and reduced reproductive autonomy [8]. The recent control of the country by the Taliban in 2021 has worsened the living condition for women [14]. The Taliban have removed the Ministry of Women’s Affairs, which promoted women’s rights through Afghan laws [14]. Additionally, women’s education and participation in government have been limited. The Taliban have also dismantled the EVAWL [14].

Studies have shown several adverse health outcomes associated with violence experience, including unintended pregnancy [15], mental disorders [16], and adverse pregnancy outcomes [17]. Violence may lead to unintended pregnancy through not using any contraception or not using contraception that are controlled by the woman, e.g., pills, patches, rings, injectables, intrauterine devices (IUDs), implants, or female sterilization [18]. Studies on the association between intimate partner violence (IPV) and contraceptive use in contexts similar or comparable to Afghanistan have shown contradicting results. Some studies have reported that women who experience IPV are less likely to use contraception and more likely to experience intrusion in their efforts of doing so [18]. Other studies have found that IPV was associated with using any contraceptive method [19]. Research shows that women with experience of IPV are more likely to endure reproductive control by their male partners, whereas male reproductive control has been associated with unintended pregnancy [20]. Therefore, to prevent unintended pregnancy, women with experience of IPV are more likely to use contraceptives that are in their control, such as pills and injectables, without the knowledge of their partner [21]. Recent research in this field shows increased contraceptive use among women with experience of IPV [21,22,23]. However, little research in low- and middle-income countries has gone beyond examining whether IPV is associated with any contraceptive use, and no study has examined whether IPV is associated with any contraceptive use in Afghanistan, despite the high prevalence of spousal physical violence (56%) and the low rate of contraceptive use (23%) in the country [12]. Therefore, it remains unclear whether spousal violence is associated with contraceptive use in Afghanistan, and whether women who experienced any type of spousal violence or an increased number of types of spousal violence are more likely to use specific types of contraceptives that are in their control. We hypothesized that Afghan women’s experience of any spousal violence would be associated with contraception use, and among those using contraception, women with experience of two or three types of spousal violence would be more likely to use female-involved methods compared to male-involved methods. Accordingly, we first examined the association between the number of types of spousal violence and any contraceptive use; second, we investigated the association between the number of types of violence and contraceptive type use among those using contraceptives. Such a study is essential in understanding Afghan women’s need for certain types of contraceptives and the continuing international efforts in addressing gender inequalities and family planning in this war-torn country. This study also has significant relevance considering the current increasing violence against women under the Taliban administration.

## 2. Materials and Methods

Data from the 2015 Afghanistan Demographic and Health Survey (AfDHS), the first cross-sectional national survey administered in Afghanistan as a part of the Demographic and Health Surveys (DHS) program, were used. The 2015 AfDHS national sample covered all 34 provinces of Afghanistan and provided up-to-date data on various subjects [12]. In this study, in accordance with our hypotheses that are based on current literature in the field, we used the modules on spousal violence, family planning (contraceptive use and desire for future pregnancies and children), and demographics.

AfDHS data collection was performed through interviews in an eight-month period from 15 June 2015 to 23 February 2016 [12]. The CSPro computer package was used for data entry, and double entry was performed for 100% verification. This simultaneous processing of the data minimized data entry errors [12].

The 2015 AfDHS utilized a stratified two-phase sampling design, and sampling weights were based on sampling probability calculated for each sampling stage and cluster; this method allowed for estimates of key indicators at the national level [12]. In the first phase, clusters consisting of enumeration areas were selected. In the second phase, a systematic sampling of households was performed, and 27 households per cluster were identified, resulting in a total of 25,650 households [12]. From the 22,132 selected women, 21,324 were successfully interviewed, yielding a response rate of 96%. About 808 women could not be interviewed due to privacy and security issues. The detailed methodology for this data set can be found in AfDHS 2015 [12].

In accordance with ethical guidelines of the World Health Organization (WHO), only one woman per household was interviewed for the domestic violence module [12]. The sample was restricted to married couples because sex before marriage is forbidden in Afghanistan; therefore, there were no data on unmarried individuals and contraceptive use [12]. Of the 21,324 participants, 18,985 were not missing on any variable included in this study. The data collection tools and procedures were approved by the Inner-City Fund (ICF) Institutional Review Board (IRB) and Afghanistan’s Ministry of Public Health [12]. We received IRB exemption from the University of Maryland IRB. This study was preregistered to access the data online, as access is only granted for legitimate research purposes.

Married participants between the ages of 15–49, and were asked if they were currently using any of the following methods to prevent pregnancy: periodic abstinence, condoms, withdrawal, pills, injectables, intrauterine device (IUD), implants, Lactation Amenorrhea Method (LAM), and female and male sterilization [12]. LAM is a contraception method where women exclusively breastfeed, which helps their body stop ovulating, and, consequently, they cannot get pregnant [12]. We had two dependent variables. The first dependent variable coded whether women reported currently using any of these methods except periodic abstinence. For those who reported using one of these methods except periodic abstinence, they were coded as using contraception. Those who reported using no method or that they were using periodic abstinence were considered using no method. We put periodic abstinence in the category of using no method because periodic abstinence is unlikely to protect against pregnancy [24], and, also, in our sample, only 0.06% (17) participants reported using periodic abstinence. The second dependent variable was the type of contraceptive method the person was using, grouped into one of six categories based on typical use effectiveness, male involvement, and amount of user behaviors required: male-involved methods (condoms, withdrawal, and male sterilization), pills, injectables, IUDs and implants, female sterilization, and LAM [24].

The number of types of spousal violence was categorized as none, one type, or two or more types based on women’s reported experiences with verbal, physical, and sexual violence in the past 12 months. For physical spousal violence, women were asked if their husbands ever pushed them or threw something at them; twisted their arm or pulled their hair; slapped them; kicked them, dragged them, or beat them; attempted to choke or burn them; or threatened or attacked them with a knife, gun, or any other weapons [12]. For sexual spousal violence, women were asked if their husbands ever physically forced or threatened them to have sexual intercourse or perform sexual acts even when the women did not want to [12]. Lastly, for emotional violence, women were asked if their husbands ever did or said something to humiliate them in front of others; threatened to harm or hurt them or the people close to them; or insulted them or made them feel bad about themselves [12]. The above binary variables were defined as yes if a woman reported experiencing any of the forms of violence from a current husband in the past 12 months.

We included the covariates of age, education, wealth, parity, residency, desire for more children, and decision-making power on contraceptive use. Age was grouped as 15–24, 25–34, and 35–49, and education was categorized as no education, primary, secondary, and higher than secondary education. The wealth index, based on household income and spending, was categorized as richest, richer, middle, poorer, and poorest. We classified the total number of children ever born into 1, 2–4, or more children (≥5). The place of residence was classified as urban and rural, and the desire for future children was dichotomized into yes or no. Finally, for those using a contraceptive method, decision-making power on contraceptive use was categorized as shared decision making, mainly respondent, and mainly husband.

First, we present univariate statistics on all variables of interest. Second, we conducted chi-squared tests to evaluate the bivariate relationships between the number of types of spousal violence experienced in the last 12 months (exposure), contraceptive use (dependent variable), and the covariates, including age, education, wealth, residency, parity, and future desired children. Third, among all women, we conducted unadjusted and adjusted logistic regression analyses to examine the association between the number of types of spousal violence experienced and any contraceptive method use. Then, among women using a method, we ran an adjusted multinomial logistic regression to examine the association between number of types of spousal violence experienced and the type of contraceptive method used. Male-involved methods were our reference group. Analyses were conducted using the statistical software, Stata 16, taking the complex survey sample design (e.g., sample weighting) into account using the svy Stata command.

## 3. Results

From a total sample size of 18,985 Afghan married women, 29.7% (5641) experienced spousal emotional violence, 43.9% (*n* = 8327) experienced spousal physical violence, and 7.4% (*n* = 1402) experienced spousal sexual violence. Table 1 presents the characteristics of the study sample according to the status of spousal violence. Overall, 21.0% (*n* = 3979) experienced one type of spousal violence, and 27.3% (*n* = 5187) experienced more than one type of spousal violence. Of those who experienced one type of spousal violence, physical violence was most likely to be experienced (79.4%). Furthermore, 23.1% (*n* = 4388) of the women used some type of contraceptive method: 5.6% (*n* = 1068) used male-involved methods (condoms, withdrawal, or male sterilization), 6.8% (*n* = 1283) used pills, 6.2% (*n* = 1178) used injectables, 1.5% (*n* = 288) used IUD or implants, 1.7% (*n* = 327) used female sterilization, and 1.3% (244) used LAM.

In bivariate analyses (Table 1), number of types of intimate partner violence was associated with contraceptive use and each covariate. Those with more than one type of spousal violence experience were more likely to use contraception than their counterparts without spousal violence experience (29.0% vs. 20.5%, *p* < 0.000). In addition, those who reported experiencing more than one type of spousal violence were more likely to have no education compared with women who did not experience any spousal violence (89.6% vs. 82.9%, *p* < 0.000). There were also significant differences with regard to the place of residence. Those who had experienced spousal physical violence were more likely to be residents of rural areas than those without the experience (79.6% vs. 70.9%, *p* < 0.000).

In both unadjusted and adjusted models examining the association between the experience of and number of types of spousal violence and any contraceptive use (Table 2), compared to women who did not experience any spousal violence, women who experienced two or three types of spousal violence had higher odds of using contraception (unadjusted odds ratio (OR) = 1.84, 95% confidence interval (CI): 1.57–2.16; adjusted odds ratio (aOR) = 1.93, 95% CI: 1.64–2.27, *p* = 0.000). Experiencing one type of spousal violence was not associated with using contraception (aOR = 1.13; 95% CI: 0.93–1.73, *p* = 0.210). Other covariates that were significantly associated with using any contraceptive method were having more education, living in an urban area, being wealthier, having five or more children, and not desiring any (more) children. 

We also examined the association of the number of types of spousal violence and the method type among the 4457 women using a contraceptive method (Table 3). After the adjustment for covariates, both women who experienced one type of violence (aRRR = 1.71, 95% CI: 1.17–2.52, *p* = 0.004) and those who experienced two or three types of spousal violence (aRRR = 2.12, 95% CI: 1.63–2.77, *p* = 0.000) were more likely to use pills than male-involved methods. In addition, women who experienced two or three types of spousal violence were more likely to use injections (aRRR = 1.75, 95% CI: 1.26–2.41, *p* = 0.001) and LAM (aRRR = 3.27, 95% CI: 2.05–5.20, *p* = 0.000) than male-involved methods.

## 4. Discussion

Our study is the first to assess the association between spousal violence and contraceptive use among a nationally representative sample of Afghan women, making our findings generalizable to Afghan women. We found that approximately one in two Afghan women experienced spousal violence in the past year, more than one in four experienced two or three types of spousal violence in the past year, and one in five used contraceptives. Of those who experienced one type of spousal violence, 79.4% experienced physical violence. We found that compared to women who did not experience any spousal violence, women who experienced two or three types of spousal violence were significantly more likely to use any contraceptive method, and, in particular, pills, injectables, and LAM, compared to male-involved methods. A potential explanation for why Afghan women who experience more types of spousal violence were more likely to use contraceptives is that Afghan women are worried about having more children who may also suffer from the violence they are experiencing. Afghan women may be particularly fearful about having a female child who may suffer from violence for being a female [25]. 

Among women using contraception, women experiencing more types of spousal violence may be more likely to choose pills, injectables, or LAM compared to male-involved methods because these methods are under the women’s control, do not involve the spouse, and may be hidden from their spouses. Though not significant, the relative risk ratio (RRR) showed that spousal violence was associated with a lower likelihood of using implants and IUDs than male-involved methods, and this may be because of misinformation about implants’ and IUDs’ side effects or lack of access; in our study, only 1.7% of the participants used implants and IUDs. Implants and IUDs are not as accessible to women as pills and other methods are [12,26]. Our findings highlight the pervasiveness of spousal violence in Afghanistan and the importance of making pills and injectables, or other methods not under control of men, more accessible to Afghan women. The findings of this study are essential to continuing international efforts in pressuring the Taliban to protect women’s rights, and in addressing gender inequalities and family planning issues in the war-torn country.

Our findings are aligned with previous studies conducted in South Asia, including India [19], Sub-Saharan Africa [22], and the U.S. [27], all of which found that having experienced intimate partner violence (IPV) was associated with using contraception or not using male-involved methods. However, our results are not consistent with other studies that have been conducted in Colombia [28], Estonia [29], and the U.S. [30] that found IPV was associated with not using contraception or using male-involved methods. In addition, a study in India by Stephenson et al. found an association between IPV and contraceptive use in two northern states (Bihar and Jharkhand), but not in two southern states (Maharashtra and Tamil Nadu) [31]. The authors suggest that the difference in findings of the states in India might be due to higher levels of women’s autonomy in the two included southern states compared to the two included northern Indian states [31]. A potential explanation for inconsistent findings among studies is that the association between spousal violence and contraceptive use depends on the type of methods available for use, as well as the cultural context. Future research that compares reasons for not using contraception and using various methods in different cultural contexts is needed.

Because spousal violence is highly prevalent in Afghanistan [12] and has been noted by both media and other studies conducted in Afghanistan [32,33], there may be less social desirability bias for reporting spousal violence than in other places. Steps were taken to minimize social desirability bias—women were asked about spousal violence in a private setting in the absence of their husbands, so that women could feel safe and report accurate information [12]. However, our study has some limitations, including a lack of data on contraception access, which hinders our understanding of why some women were not using contraception. We also do not have information about why women chose the methods they did, nor the socio-cultural issues. In this study, spousal violence was a binary variable, which inhibited our ability to account for the gravity of each type of spousal violence that may influence the use of contraception differently. We were also unable to account for other confounders, including accessibility of contraception, experience of violence (prior to the 12 months), and insurance or cost of contraceptives, that may impact the association between spousal violence and contraceptive use. Due to the ongoing conflict and war in Afghanistan, data collection is very challenging. The 2015 AfDHS data used in this study are from the first survey ever performed at the national level in Afghanistan. Although our data are from 2015 and there have been political changes in Afghanistan, it is very likely that our findings are still relevant, since women’s conditions have not improved in the country.

## 5. Conclusions

Afghanistan is one of the countries with the highest prevalence of spousal violence. Our findings suggest that spousal violence is associated with any contraceptive use, and, in particular, pills, injectables, and LAM, compared to male-involved methods in Afghanistan. The findings of this study may inform policymakers and program implementers in designing interventions to address the pervasive problem of violence against women, and make pills and injectables more accessible to Afghan women, since these methods are under women’s control and more often used in Afghanistan. Future research on the socio-cultural issues surrounding spousal violence and contraceptive use in Afghanistan is important to better understand why there is an association between spousal violence and any contraceptive use. Future studies should also explore the role of campaigns in promoting women’s rights and contraception education. Research shows that previous campaigns on gender-based violence in Afghanistan were effective [34].

## Figures and Tables

**Table 1 ijerph-19-09783-t001:** Descriptive statistics of the sample of Afghan married women aged 15–49; AfDHS 2015.

		Number of Types of Spousal Violence in the Last 12 Months			
	*n* (%)	*n* (%)	*n* (%)	*n* (%)	
Variables	Total, 18,985	None, 9819 (51.7%)	1 Type, 3979 (21.0%)	2 or 3 Types, 5187 (27.3%)	*p*-Value
**Contraceptive use**					<0.000
No	14,570 (76.9)	7794 (79.5)	3099 (78.0)	3677 (71.0)	
Yes	4388 (23.1)	2012 (20.5)	876 (22.0)	1500 (29.0)	
**Method Types**					<0.000
None	14570 (76.9)	7794 (79.5)	3099 (78.0)	3677 (71.0)	
* Male-involved methods	1068 (5.6)	535 (5.5)	223 (5.61)	310 (6.0)	
Pills	1283 (6.8)	539 (5.5)	272 (6.84)	472 (9.1)	
Injectables	1178 (6.2)	546 (5.6)	210 (5.28)	422 (8.2)	
IUD/implants	288 (1.5)	155 (1.6)	62 (1.56)	71 (1.4)	
Female sterilization	327 (1.7)	165 (1.7)	64 (1.61)	98 (1.9)	
LAM	244 (1.3)	72 (0.7)	45 (1.13)	127 (2.5)	
**Age**					<0.000
15–24	3730 (19.7)	2073 (21.1)	726 (18.3)	931 (18.0)	
25–34	7713 (40.6)	3852 (39.2)	1643 (41.3)	2218 (42.8)	
35–49	7542 (39.7)	3894 (39.7)	1610 (40.5)	2038 (39.3)	
**Education**					<0.000
No education	16309 (85.9)	8140 (82.9)	3520 (88.5)	4649 (89.6)	
Primary	1276 (6.7)	726 (7.4)	269 (6.8)	281 (5.4)	
Secondary	1091 (5.8)	721 (7.3)	159 (4.0)	211 (4.1)	
Higher	309 (1.6)	232 (2.4)	31 (0.8)	46 (0.9)	
**Residency**					<0.000
Urban	4741 (25.0)	2862 (29.2)	823 (20.7)	1056 (20.4)	
Rural	14244 (75.0)	6957 (70.9)	3156 (79.3)	4131 (79.6)	
**Wealth**					<0.000
Poorest	3765 (19.8)	1918 (19.5)	743 (18.7)	1104 (21.1)	
Poorer	4265 (22.5)	2141 (21.8)	961 (24.2)	1163 (22.4)	
Middle	4054 (21.4)	1899 (19.3)	960 (24.1)	1195 (23.0)	
Richer	3990 (21.0)	2064 (21.0)	847 (21.3)	1079 (20.8)	
Richest	2911 (15.3)	1797 (18.3)	468 (11.8)	646 (12.5)	
**Parity**					<0.000
1	1954 (10.3)	1131 (11.5)	364 (9.2)	459 (8.9)	
2–4	7173 (37.8)	3681 (37.5)	1466 (36.8)	2026 (39.1)	
≥5	9858 (51.9)	5007 (51.0)	2149 (54.0)	2702 (52.1)	
**Desire for more or any children**					<0.000
No	11,981 (63.1)	6481 (66.0)	2474 (62.2)	3026 (58.3)	
Yes	7004 (36.9)	3338 (34.0)	1505 (37.8)	2161 (41.7)	

* Male-involved methods: condoms, withdrawal, and male sterilization.

**Table 2 ijerph-19-09783-t002:** Logistic regression model presenting the association of spousal violence with any type of contraceptive use among married Afghan women (*n* = 18,985); AfDHS 2015.

Variables	Unadjusted Odds Ratio UOR [CI]	*p*-Value	Adjusted Odds Ratio AOR [CI]	*p*-Value
**Spousal Physical Violence**				
None			Ref.	
1 type	1.15 [0.94–1.39]	<0.074	1.13 [0.93–1.73]	<0.210
2 or 3 types	1.84 *** [1.57–2.16]	<0.000	1.93 *** [1.64–2.27]	<0.000
**Age**				
15–24			Ref.	
25–34			0.83 [0.65–1.06]	<0.135
35–49			0.92 [0.69–1.21]	<0.529
**Education**				
No education			Ref.	
Primary			1.45 ** [1.14–1.84]	<0.002
Secondary			1.77 ** [1.23–2.56]	<0.002
Higher			2.37 * [1.22–4.60]	<0.011
**Residency**				
Urban			Ref.	
Rural			0.72 * [0.56–0.93]	<0.012
**Wealth**				
Poorest			Ref.	
Poorer			1.18 [0.98–1.41]	<0.074
Middle			1.14 [ 0.84–1.56]	<0.405
Richer			1.79 *** [1.35–2.38]	<0.000
Richest			2.41 *** [1.71–3.59]	<0.000
**Parity**				
1			Ref.	
2–4			1.66 *** [1.29–2.15]	<0.000
≥5			2.61 *** [1.89–3.59]	<0.000
**Desire for more or any children**				
No			Ref.	
Yes			0.84 * [0.71–0.99]	<0.041

REF indicates the referent group. *** *p*-value < 0.05, ** *p*-value < 0.01, * *p*-value < 0.1. UOR stands for unadjusted odds ratio, AOR stands for adjusted odds ratio, and CI is the confidence interval.

**Table 3 ijerph-19-09783-t003:** Association of spousal violence with methods by effectiveness level among women using contraception.

	Methods by Effectiveness Level (Base Outcome: Male-Involved Methods)				
Variables	Pills	Injectables	IUD & Implants	Female Sterilization	LAM
**Spousal Physical Violence**					
None	Ref.	Ref.	Ref.	Ref.	Ref.
1 type	1.71 [1.17–2.52]	1.16 [0.70–1.93]	0.63 [0.30–1.33]	1.20 [0.69–2.07]	1.10 [0.57–2.12]
2 or 3 types	2.12 [1.63–2.77]	1.75 [1.26–2.41]	0.71 [0.40–1.24]	0.87 [0.59–1.29]	3.27 [2.05–5.20]
**Age**					
15–24	Ref.	Ref.	Ref.	Ref.	Ref.
25–34	0.68 [0.42–1.09]	1.19 [0.71–2.01]	0.94 [0.41–2.16]	49.22 [6.19–391.61]	0.71 [0.39–1.30]
35–49	0.51 [0.31–0.84]	1.48 [0.84–2.62]	1.28 [0.41–3.99]	238.0 [28.59–1981.8]	0.24 [0.10–0.57]
**Education**					
No education	Ref.	Ref.	Ref.	Ref.	Ref.
Primary	0.51 [0.34–0.76]	0.63 [0.39–1.00]	0.82 [0.40–1.69]	0.79 [0.40–1.53]	0.26 [0.09–0.73]
Secondary	0.67 [0.39–1.16]	0.48 [0.27–0.87]	1.59 [0.79–3.24]	0.51 [0.22–1.20]	0.12 [0.03–0.45]
Higher	0.19 [0.07–0.53]	0.97 [0.42–2.24]	3.23 [1.04–10.00]	0.10 [0.02–0.56]	0.40 [0.03–4.62]
**Residency**					
Urban	Ref.	Ref.	Ref.	Ref.	Ref.
Rural	1.64 [1.06–2.53]	1.57 [0.92–2.68]	2.52 [0.79–8.05]	1.14 [0.67–1.94]	1.01 [0.38–2.66]
**Wealth**					
Poorest	Ref.	Ref.	Ref.	Ref.	Ref.
Poorer	0.54 [0.31–0.95]	0.49 [0.28–0.84]	2.32 [0.97–5.51]	0.51 [0.22–1.19]	1.88 [0.56–6.37]
Middle	0.44 [0.25–0.77]	0.39 [0.22–0.69]	1.03 [0.46–2.32]	0.46 [0.20–1.08]	1.33 [0.41–4.37]
Richer	0.39 [0.23–0.65]	0.29 [0.16–0.52]	1.65 [0.71–3.83]	0.42 [0.20–0.92]	0.82 [0.26–2.65]
Richest	0.53 [0.28–1.00]	0.21 [0.11–0.41]	2.12 [0.59–7.55]	0.59 [0.24–1.46]	0.54 [0.13 –2.32]
**Parity**					
1	Ref.	Ref.	Ref.	Ref.	Ref.
2–4	1.94 [1.11–3.39]	3.96 [1.63–9.65]	3.56 [1.01–12.47]	3.42 [0.43–27.42]	1.26 [0.63–2.49]
≥5	2.17 [1.18–3.98]	5.21 [1.83–14.86]	3.30 [0.77–14.11]	6.38 [0.76–53.22]	1.52 [0.63–3.68]
**Desire for more or any children**					
No	Ref.	Ref.	Ref.	Ref.	Ref.
Yes	1.00 [0.75–1.33]	0.63 [0.46–0.86]	0.93 [0.54–1.60]	-	2.04 [1.14–3.65]
**Decision-making**				-	
Shared decision making	Ref.	Ref.	Ref.	Ref.	Ref.
Mainly respondent	1.61 [1.05–2.46]	1.03 [0.65–1.64]	1.24 [0.67–2.28]	1.03 [0.54–1.96]	2.82 [1.47–5.40]
Mainly husband	0.52 [0.33–0.80]	0.60 [0.38–0.94]	1.20 [0.63–2.28]	0.67 [0.35–1.26]	4.08 [1.51–6.63]

Data are adjusted risk ratio [95% confidence interval]. Ref., the reference group.

## Data Availability

The publicly available data were obtained from DHS website at: https://dhsprogram.com/data/dataset/Afghanistan_Standard-DHS_2015.cfm?flag=0 (accessed on 8 December 2019).

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
