# Peer review of "Spousal Violence and Contraceptive Use among Married Afghan Women in a Nationally Representative Sample"

_ijerph, 2022, doi:10.3390/ijerph19169783_

Round 1

Reviewer 1 Report

Abstract

The present study aimed to explore the possible relation between violence (both in terms of

presence/absence and in terms of different kinds of violence perpetrated) against spouses and

contraception use (and eventually the types of methods adopted), among the Afghan population. In

order to achieve this, the authors have analyzed data from the 2015 Afghanistan Demographic and

Health Survey, which investigated the condition of 18,985 Afghan married women (ages 15 to 49).

The study results reported how women who experienced intimate partner violence were more likely

to use contraception methods that were directly under their control (for example the contraceptive

pill) instead of male-controlled kinds of contraception. The authors hypothesize that these results

might be linked to the fact that women who experience this kind of violence might not trust at all

their partner and therefore might desire a contraceptive method that can help them become fully in

control of their reproductive health. This can have implication for practice, especially considering

the intense need to provide these women with reliable contraceptive methods they can fully manage

by themselves, which are still scarcely used in the country.

Authors’ abstract

The abstract provided by the authors contains all the general information typically required to

summarize the study, however, it does not manage to fully convey the fundamental reasons that led

the authors to investigate the described phenomenon (that is, the eventual relation between the

presence of spousal violence and the type and use of contraceptives among Afghan women).

Moreover, the same kind of shortcoming can be detected concerning the reported results, which are

presented in a way that might not be perceived as engaging by readers. Since the abstract is the very

first element that readers encounter in an article, it would be advisable to add few, very concise

sentences, which could better explain why the authors felt important to investigate the described

variables in the first place and what kind of meaning and contribution to research their findings can

offer. As has already been indicated, these modifications do not need to be extensive, since that

would be beyond the scope of an abstract, but they would nonetheless improve the first impact the

reader can have with the present research.

Introduction

In this first section of the article, the authors provide a description of the theoretical background in

which their research is set, presenting some of the sources in the scientific literature related to the

theme of intimate partner violence in Afghanistan and the use of contraception in the country,

together with other international studies that have explored the possible relation between the two

variables.

Overall, the Introduction section of the present article appears to be well-written and organized, and

the authors generally present an adequate description of the current literature, with studies that

appear to be mostly recent, a very important element to consider when writing an article.

However, the section would also benefit from a revision, since there are some aspects that should be

broadened and described more in depth.

More specifically, on page n. 3, at the beginning of the Introduction section, the authors state:

“Conflict also robs people of opportunities for education and employment, leading to poverty,

inequality, oppression, and stress, all of which can induce and exacerbate domestic violence. Men’s

diminished opportunities for employment and providing for their families may sabotage their self-

evaluation as providers, causing stress and anxiety, which can be expressed in the form of spousal

violence”.

Since this aspect represents a crucial one for the present study, the authors should consider the

possibility to explain more in depth what are the variables and mechanisms through which poverty,

inequality, oppression, stress and anxiety can induce and exacerbate domestic violence. The theme

is a very interesting and complex one, since of course not all situations that include the previously

mentioned variables lead to domestic violence, therefore, it would be useful if the authors could

dedicate some more time to this aspect, and explicate, basing on current literature, in which

conditions and contexts these elements can indeed result, or have a higher risk of resulting, in

domestic violence. This would, moreover, contrast the subtle justification and victimization of men

acting out domestic violence and open the field for introducing important background information

about the social and cultural norms that lead to the phenomenon. The authors should consider

therefore to draw a connection between the above-mentioned statement and literature on hegemonic

masculinity, patriarchy, and sexism. Otherwise this paragraph could risk to suggest a causal

influence between war and domestic violence which would represent an oversimplification of the

more complex social and cultural dynamics in Intimate Partner Violence (IPV).

Moreover, in relation to the previous aspect, the authors, few lines after, also state: “In Afghanistan,

there are also cultural and legal factors that deprive women of rights and privileges granted to men

only”. Since this represents a fundamental aspect as well for the understanding of the social and

cultural background in which the study is inserted, the authors should consider the possibility to

broaden the description of such aforementioned cultural and legal factors that discriminate between

men and women in Afghanistan. The authors do mention some of these elements in the subsequent

lines of the manuscript, however, a more in-depth description of them would prove to be

particularly informative especially for international readers who might not be completely familiar

with Afghanistan laws and culture.

Moreover, it is to be considered positively that the authors refer to the latest developments in

Afghanistan under Taliban government. Nonetheless, as the data used in the present article is from

2015, it would be useful for the reader to understand more about the Afghan society in 2015, before

the control of the Taliban regime.

Furthermore, still on page n. 3, the authors also mention the possible negative outcomes domestic

violence can have on women (“Studies have shown several adverse health outcomes associated with

violence experience, including unintended pregnancy, mental disorders, and adverse pregnancy

outcomes”). However, since these elements represent a very important aspect that needs to be

intensely highlighted when discussing about domestic violence, the authors should consider the

possibility to broaden their description some more, possibly adding examples from other studies in

literature that have detected and explored these outcomes.

Another aspect that would benefit from a revision can be found on page n. 4, when the authors

explore some studies that have reported a correlation between domestic violence and contraceptive

use. More specifically, they indicate that “Other studies have found that IPV was associated with

using any contraceptive method”. Since such a correlation might not appear to be completely clear

and could even seem confusing to readers, the authors should explain more in depth the reasons (or

hypotheses) these studies have found to substantiate such a correlation. This aspect is indeed very

interesting and possibly not commonly studied, and it represents the fundamental element that leads

the entire present research, therefore readers could certainly benefit from receiving a more in depth

description of what is currently known concerning this phenomenon.

Lastly, another aspect the authors should revise is the fact that, at the end of the Introduction

section, they present the general objectives of their study, that is, to examine the possible presence

of a relation between spousal violence and contraceptive use, however, they do not describe in

depth their eventual study hypotheses, in case they had formulated some specific ones. Moreover,

since, as previously noted, the Introduction section does not fully describe the relation that over the

years has been hypothesized and observed between these variables, the reason that led the authors to

explore these aspects does not appear completely clear in turn, therefore, the authors should broaden

this part of the Introduction, for example by inserting more examples from the current literature

concerning the detected relation between spousal violence and contraceptive use.

This would allow readers to better understand the basis on which the present research has been

designed.

Materials and Methods

In this section the authors describe the kind of methodology and the procedures they adopted for the

implementation of the research and for data analysis.

Although the section is generally well-written, there are some aspects that require a revision.

First of all, as has already been highlighted, it appears unclear exactly what kind of hypotheses the

authors have followed for the implementation of their research, a shortcoming whose consequences

can also and mostly be detected in this section, given that the authors' way of proceeding is not

particularly clear. Therefore, the authors should indicate whether they had already formulated

specific hypotheses on the basis of current literature in the field, and how these hypotheses guided

the implementation of their study.

The description of the dependent variable seems to be sufficiently clear, however, the authors

should consider adding a brief explanation for the choice to categorize periodic abstinence as use of

no contraception, as this aspect could not be immediately clear to all readers. Moreover, the authors

use the acronym LAM, without further describing what kind of procedure it represents. Since, as

already noted, not all readers might be experts in quantitative research, the authors should better

explain what LAM means.

Furthermore, at the end of this section, the authors provide some indications about the program used

for the analysis, however the sentence: "Analyzes were conducted using Stata 16, taking the

complex sample design into account using the svy command" appears to be rather unclear,

especially for a reader who has no in-depth knowledge of the field. Therefore, the authors should

consider the possibility to explain more clearly the meaning of the sentence and describe the steps

related to the data analysis procedures with a language that can be friendly even for people who are

interested in the study but not experts in statistical programs, in order to make the article accessible

as much as possible for any kind of professional.

Moreover, the authors also refer to conducting both chi-square tests and logistic regression,

however, is not immediately clear what additional information can be drawn from this operation.

Lastly, with regards to ethical issues, the authors should add a brief paragraph on the possible

implications of investigating such a delicate topic. The approval number of ICF and Afghanistan’s

Ministry of Public Health are lacking. Moreover, information is lacking concerning whether the

study has been preregistered.

Results

In this section the results obtained from the data analysis are described and the statistics used for the

study of the variables considered and their correlations are presented.

This part of the manuscript appears generally clear and well-structured.

However, some aspects need a precise and attentive revision. For example, on page 8, references to

the main results of bivariate analysis is lacking (chi square, degrees of freedom etc.), since only

reporting the p-value is not considered to be sufficient.

Moreover, the authors should also add a more precise description of which kind of bivariate

analysis has been conducted.

Furthermore, according to the guidelines of the Cochrane manual on statistical presentation, p-

values lower than 0.001 should be reported as p< 0.001, reporting thus only up to 3 decimal places.

Therefore, the authors should correct data presented in Table 1 accordingly, since some results are

presented with 1 and other with 2 decimal places. Moreover, relevant test statistics should be added

to the table as reporting p-values only is not sufficient. The test statistics on wealth seem to be

lacking.

The authors should also check upon font type and font size throughout the Result section as there

seem to be some inconsistencies.

Furthermore, on page 9, the authors should add the p-value to all reports on OR test statistics

considering again the indications of Cochrane with respect to decimal places.

Another aspect that needs to be revised is the fact that in Table 2, the authors should add a brief

explanation regarding which numbers refer to OR and which to CI. Moreover, they should also add

an explanation for the use of asterisks. This information could be intuitively clear but need

specification to enhance readability.

Moreover, with regard to the multinominal logistic regression, a lot of statistical reporting seems to

be lacking (χ2, degrees of freedom, N, p-value, pseudo R²), which is needed before moving to

reporting odds ratio. This aspect of the Results section requires some fundamental and extensive

review.

Lastly, the authors should try to enhance the readability of Table 3, eventually by adding asterisks.

Discussion

In this section the authors restate the research objectives and results, comparing them with some of

the main sources that can be found in literature concerning the considered theme.

A commendable aspect of this part of the article is that the explanation of the obtained results that

has been provided appears generally adequate and consistent with the objective data retrieved.

However, there are some elements that would benefit from a careful revision.

For example, on page 12, it could be useful to discuss more closely the aspects directly related with

the present research before broadening the view to causal hypothesis on underlying reasons for the

association between IPV and contraception.

Moreover, on page 13, the authors should introduce the abbreviation “RRR” before using it in the

text as it may be hard for the reader to understand to what it refers.

It could be also useful to reflect upon implications, similarities and differences between the

concepts of spousal violence and IPV as well as the role of hegemonic masculinity, patriarchy and

sexism in the self-determination of contraception in women.

It would be moreover useful to indicate studies in literature that can support the thesis that there

might be lower odds of using implants and IUDs because of misinformation or lack of access to it

among Afghan women.

Moreover, the authors should discuss more in depth some important limitations of the present study,

therefore it could be useful adding a limitations section. First, the dataset used seems to be quite

robust, justifying the authors implications to be quite representative of Afghan society. Nonetheless,

the authors should consider adding an explanation concerning the choice to use a rather old dataset,

of about 7 years ago. In a country which most likely is undergoing rapid changes in government and

war activity this time gap could be an important limitation for inferences for the current situation.

Moreover, as answers to an interview cannot considered to be completely independent from the

social circumstances, culture and power structures of the interview setting, especially for topics as

sensitive as IPV and contraception, it would be important to report if the survey was enrolled by

local vs. foreign and male vs. female experimenters and possible implications of this circumstance.

Lastly, the authors should consider discussing the limitation of the IPV variable. Considering only a

quantitative binary aspect of IPV presence, without taking into account the gravity of each type of

IPV. Gravity of IPV could be an important factor for use of contraception, so that the lack of this

information in the dataset should be described as a limitation. Also, past experiences of IPV (more

than 12 months ago) could be taken into account as a confounding variable that has not been

investigated. Finally, given the circumstances of war, information is lacking if widows were

included in the study and if some women’s’ husbands were absent for combat (leading to a lack of

need for contraception, eventually).

Authors should also consider to broaden the discussion with implications of the present study that

go beyond a more in-depth understanding of the Afghan context and contribute to the general

dispute on the relationship between contraception and IPV. They should discuss further possible

underlying covariates or factors that influence the relationship between contraception and IPV that

the study could not take into account.

Conclusion

In the Conclusion section, the authors summarize their study objectives and main findings. A

commendable aspect is that the authors highlight the possible implications for practice that could

derive in terms of new policies and interventions concerning violence against women and the

necessity to render contraceptives more available.

The overall relevance of the present study, both for research and policy interventions in Afghanistan

and for a further understanding of the phenomenon of spousal violence and contraception in non-

western countries is to be considered very high. The authors provide an adequate outlook for the

implementation of the present results in the Afghan context, promoting women’s rights and self-

determination by public health interventions.

References

This section appears mostly adequate, since it faithfully reports all the references mentioned in the

manuscript.

However, the format in which the references have been written does not appear completely in line

with the journal recommendations (for example, the authors did not insert a “; between the names of

different authors, they did not always write the name of journals in italics etc.), therefore the

authors should carefully revise the section, checking every reference in order to correct the small

inaccuracies.

Specific Areas to be Considered as Required by the Journal

Since the Journal also requires reviewers to answer to some specific questions and evaluate some

precise elements of the manuscript, even though the general answer to such questions is already

present in the review report, nonetheless in the following section all the fundamental areas to be

considered have been reported and for each of them the reviewer’s precise opinion has been added.

Novelty: Is the question original and well-defined? Do the results provide an advancement of the

current knowledge?

The study question appears to be original and interesting, and the authors report no previous studies

had been conducted in their country concerning the relation between spousal violence and

contraception use. However, the study question does not appear to be completely well-defined. The

authors, as has already been highlighted, should indeed better explain the reasons that led them to

consider the indicated variables and describe, if present, the general hypotheses they had formulated

before the implementation of the study. However, even considering the aforementioned

shortcomings, the results do represent an advancement of the current knowledge in the filed, which

is a rather unexplored area especially in Countries like Afghanistan.

Scope: Does the work fit the journal scope*?

The manuscript appears in line with the journal scope, since it explores some fundamental issues in

the field of public health and of the well being of social groups who are exposed to mayor risk of

discrimination, in particular women in countries with high rates of gender-based and intimate

violence with low levels of self-determination in society.

Significance: Are the results interpreted appropriately? Are they significant? Are all conclusions

justified and supported by the results? Are hypotheses carefully identified as such?

The study results appear to be generally appropriately interpreted, since the authors recognize that a

statistical correlation does not equal a causal relationship, and simply offer their own interpretation

of the findings in a hypothetical manner. The results also appear to be significant, even though there

are some important shortcomings regarding reporting statistical analysis which are resulting

incomplete. The conclusions that are drawn are supported by said results but should be reflected

upon more critically with regards to their limitations.

Quality: Is the article written in an appropriate way? Are the data and analyses presented

appropriately? Are the highest standards for presentation of the results used?

The article in well-written and organized, unfortunately analyses are mostly presented in an

inappropriate way.

Scientific Soundness: Is the study correctly designed and technically sound? Are the analyses

performed with the highest technical standards? Is the data robust enough to draw conclusions?

Are the methods, tools, software, and reagents described with sufficient details to allow another

researcher to reproduce the results? Is the raw data available and correct (where applicable)?

The used data is very robust and representative of Afghan women before Taliban government took

control, which allows some important inferences on Afghan society and the relationship between

spousal violence and contraception. However, research methodology, especially concerning

hypotheses and aims as well as the used statistical analysis needs some substantial revision.

Interest to the Readers: Are the conclusions interesting for the readership of the journal? Will

the paper attract a wide readership, or be of interest only to a limited number of people? (Please

see the Aims and Scope of the journal.)

The article general theme and results appear to be of great interest for a general audience, and not

only to a limited number of people, since the field of intimate partner violence and its relationship

with contraceptive use among married women is of huge importance on international level.

Therefore, the results represent an important first step in order to understand more closely this

phenomenon in Afghanistan and to guide future policies and public health interventions.

Overall Merit: Is there an overall benefit to publishing this work? Does the work advance the

current knowledge? Do the authors address an important long-standing question with smart

experiments? Do the authors present a negative result of a valid scientific hypothesis?

Because of the previously reported reasons, the publication of the present article could indeed

represent a benefit for the journal, even though the authors, as previously expressed, would need to

revise some sections of the manuscript, in particular the study objectives and methodology adopted.

However, the work does represent ad advancement in the current knowledge of the investigated

phenomenon.

English Level: Is the English language appropriate and understandable?

The English level used by the authors is absolutely appropriate and completely understandable.

Overview

The present study is to be considered of high relevance and interest to a broad public, as it

investigates a highly under-researched context.

However, there are some aspects the authors should necessarily revise.

In particular, the theoretical background of the roots of spousal violence in hegemonic masculinity,

patriarchy, and sexism as well as a more in-depth view on the state of research on the link between

IPV and use of contraception. This would lead to a clearer exposition of the research hypotheses

and main aims of the study, which are fundamental for the understanding of the methodological

choices.

Moreover, regarding reporting statistical analysis, there seem to be some very important

shortcomings as well, which necessarily need to be revised.

Lastly, the authors should discuss more in depth some important limitations of the present study.

They should also recheck the “References” section, since it presents some shortcomings.

In conclusion, the present article appears overall well-written and adequately elaborated, however a

general revision would be certainly necessary in order to address the various weaknesses that are

present.

Therefore, the general suggestion that can be given, in accordance with the Journal indications for

reviewers, is: Reconsider after Major Revisions (the acceptance of the manuscript would depend

on the revisions. The author needs to provide a point-by-point response or provide a rebuttal if some

of the reviewer’s comments cannot be revised. Usually, only one round of major revisions is

allowed. Authors will be asked to resubmit the revised paper within ten days and the revised version

will be returned to the reviewer for further comments).

Author Response

Thank you for your time and consideration. Please see the attachment for the responses.

Reviewer 2 Report

Well done, a few  minor questions attached.

Author Response

(The authors gave the same response as above.)

Reviewer 3 Report

I would like to thank the editors of this journal for the opportunity to give my point of view on this interesting research project. On the other hand, I would like to congratulate the authors of this research, I believe that research on gender violence is very necessary and all studies help with their results to create new methodologies to address this scourge and to design new proposals to help people who suffer gender violence by focusing on the countries with more difficulties to address the problem.

I believe that the study is well thought out and the methodology is appropriate, but I would like to make some comments that I believe could be addressed by the authors and improve some aspects of the study.

The size of the different sections is very short, I believe that the journal indicates a minimum of 3500 words for this type of publication (article). The introduction would benefit from discussing the campaigns that have been carried out in Afghanistan on contraception (if any) and the measures taken to curb gender-based violence (if any). Also whether these campaigns have been adopted by Afghan governments or have been developed by third parties.

I suggest reviewing the following studies:
Effective gender-based violence screening tools for use in primary health care settings in Afghanistan and Pakistan: a systematic review.

Narrative storytelling as mental health support for women experiencing gender-based violence in Afghanistan

The discussion would also benefit from an analysis of how this data can be used in the future to address the issue of gender-based violence and how it can help in the design of campaigns to promote contraception.

What works to prevent violence against children in Afghanistan? Findings of an interrupted time series evaluation of a school-based peace education and community social norms change intervention in Afghanistan

No ethics committee approval is attached or indicated in the study and no indication is given as to whether it follows the Helsinki declaration standards, as the study works with subject data.

The study addresses gender violence in its face-to-face and interpersonal dimension, but the study does not consider other non-face-to-face channels such as social networks, which I understand is not the main topic, but would help to complement future proposals. This can be seen as a limitation of the study, as it can be a good justification for future studies and to be taken into account in planning educational proposals to address the problem. Here are some references to address this in the discussion

DOI: 10.5354/0719-1529.2015.32375
http://hdl.handle.net/11162/160074
https://doi.org/10.51698/aloma.2021.39.1.27-35

As for the limitations of the study, the authors mention them in the discussion section, but I think it would be interesting to dedicate a specific section within the discussion section and better argue the limitations of the study.

Thank you again for the opportunity to evaluate this valuable research.

Author Response

(The authors gave the same response as above.)

Reviewer 4 Report

Thank you for the invitation to review this draft.Despite the previous international efforts to improve womens condition in Afghanistan, domestic violence remains a major public health concern. In this context,it is of great practical significance to discuss the current increasing violence against women under the Taliban administration.

However, I believe the paper can be strengthened furthermore if the study addresses some issues as follows:

Keywords:

I suggest replacing AfghanistanwithMarried Afghan womenis more relevant to the research topic of the article.

Materials and Methods

1.I think this sentence Married participants between the ages of 15-49 were asked if they were currently using any of the following methods”(Page 2, Line 94-95)should be mend with Married participants between the ages of 15-49 were asked if they were currently using any of the following methods in order to contraceptive,which can make the sentence more serious.

2. (Page 3, Line 108-118)I suggest that the description of these three parts be more concise, highlighting the characteristics of each storm type.

3. (Page 3, Line 119-128)It might be easier to understand if the second part of the description corresponds to the order of the words in the first sentence.

Results

(Page 3, Line 145) I suggest the authors use “21.0%” instead of “21% so that it can be consistent with the whole paragraph,and it is used in most scientific papers to reserve two decimal places.

Discussion

1.(Page 7, Line 184)different cultural contexts is needed. First, the first letter should be capitalized. Secondly, I can't understand the function   of this sentence here.

2.In the discussion section, the discussion content is not comprehensive.Therefore, this discussion section should be based on the core content of the results section of the paper.

Conclusion

(Page 7, Line 210-213)The last sentence of the conclusion can be put into the discussion section as a limitation of this paper.

Author Response

(The authors gave the same response as above.)

Round 2

Reviewer 1 Report

Dear Authors, congratulations indeed on this very interesting work! 

Reviewer 3 Report

The authors have made a substantial change to the structure of the paper and have taken all my suggestions into account. I consider the article to be suitable for publication.